# Learning Mixed Strategies in Quantum Games with Imperfect Information

Agustin Silva *, Omar Gustavo Zabaleta and Constancio Miguel Arizmendi

Instituto de Investigaciones Científicas y Tecnológicas en Electrónica (ICYTE), Av. Juan B. Justo 4302, Mar del Plata B7608, Argentina
* Correspondence: agustinsilva447@gmail.com

**Abstract:** The quantization of games expand the players strategy space, allowing the emergence of more equilibriums. However, finding these equilibriums is difficult, especially if players are allowed to use mixed strategies. The size of the exploration space expands so much for quantum games that makes far harder to find the player's best strategy. In this work, we propose a method to learn and visualize mixed quantum strategies and compare them with their classical counterpart. In our model, players do not know in advance which game they are playing (pay-off matrix) neither the action selected nor the reward obtained by their competitors at each step, they only learn from an individual feedback reward signal. In addition, we study both the influence of entanglement and noise on the performance of various quantum games.

**Keywords:** quantum computing; game theory; machine learning

## 1. Introduction

Quantum games have merged the properties of quantum mechanics with the formalism of game theory. Since its first formulation at the beginning of this millennium [1], physicists and mathematicians have developed a robust framework for describing how players behave when games are quantized [2]. Initially, changes in the equilibriums of many games were studied and it was observed how in many cases the rewards of quantum players in equilibrium surpass their classical counterparts [3]. Then, quantum games were extended from two to N players [4] and from two to M strategies [5]. Additionally, most types of games have also been quantized along with a description of their consequences [6,7]. Furthermore, the advantages of quantum games have also been used to model different scenarios from economics [8,9] to systems sharing limited resources [10,11] to network routing design [12,13].

The idea of learning in games is also becoming more and more relevant due to its applicability in scenarios, such as algorithms that learn from different self-interested data sources, economic systems that are optimized via machine learning, or even machine learning algorithms that work to find equilibria instead of minimizing a function. More broadly, the world is moving toward the coexistence of multiple AIs that learn from their interaction, which might be collaborative, strategic, or adversarial [14].

When learning in games, the way individual agents use observations to change their behavior should be explicitly specified. The design should also be such that agents play out of self-interest and not by trying to find an equilibrium [15]. There are already long-established learning strategies in the field, such as fictitious play and no-regret learning. Fictitious play is a model-base algorithm where each player chooses the best response to their opponents' average strategy, therefore players need to know the actions selected by their opponents [16]. No-regret learning, on the other hand, is a model-free approach that is based on the idea of minimizing the difference between the forecaster's accumulated loss and that of an expert, so it requires players to know the reward function or payoff

matrix [17]. That said, we propose an algorithm to learn mixed strategies in quantum games where agents only have access to one feedback reward signal. Agents adjust their strategies without information about the payoff matrix of the game they are playing, the strategies selected by their opponents nor their rewards at each step, and even how many players they are playing against. The algorithm could also be easily adapted to be applied in systems where it is necessary to learn across games, that is, in situations where agents are playing different games that share an equivalent structure at the same time [18].

The implementation of the players' strategies in the quantum device will be done using parameterized quantum circuits (PQC). PQC are a type of hybrid algorithm, part classical part quantum, which are usually performed by selecting a fixed circuit structure, parameterizing it using rotation gates, then iteratively updating the parameters to minimize an objective function estimated from measurements [19]. PQC have also been one of the main methods to integrate quantum computing and machine learning [20] since they offer a concrete way to implement algorithms and potentially demonstrate quantum supremacy in the noisy intermediate-scale quantum (NISQ) era, where computers are neither fault-tolerant to noise nor big enough to be useful [21]. Moreover, PQC were also used to calculate policies in reinforcement learning algorithm with a single agent interacting with an environment [22].

In the classical realm, there is an extensive bibliography exploring the consequences of agents learning in environments modeled using game theory. On the other hand, there have been little investigation on multiple agents learning strategies in quantum games, most of them coming from the field of quantum evolutionary game theory [23,24], which focus on pure strategies. To the best of the authors' knowledge, this is the first time that a decentralized algorithm for learning mixed strategies in quantum games has been proposed. In terms of results, the main contributions of this work are: (a) designing an algorithm to find equilibria in games with mixed quantum strategies, (b) detecting that the equilibrium strategies not only depend on the game (e.g., Prisoner's Dilemma) but also on the relation between the values of the payoff matrix, (c) verifying that entanglement is the only resource that allows games to be quantum, and (d) characterizing the sensitivity of the algorithm to find equilibriums against noise in the quantum channels.

Finally, the same motivations that drove the investigation of reinforcement learning algorithms in situations modeled by classical game theory, where players are unaware of the complete structure of the game, such as finance [25] and network routing [26], encourage us to propose this algorithm for learning in quantum games. If the future of quantum computing allows us to create networks where quantum markets and the quantum internet become a reality, decentralized algorithms which support working with mixed quantum strategy and incomplete information will be absolutely useful for individuals to make the most out of the advantages of quantum systems [27].

The rest of the work is organized as follows. In Section 2, a general description of classical and quantum games is presented. In Section 3, our learning algorithm is defined along with a detailed description of the entire model. In Section 4, the results of the application of the algorithm to classical and quantum games are presented and analyzed. In Sections 4.2 and 4.3, we include a study of the consequences of working with non-ideal entanglement and noise, respectively. Finally, the work is concluded in Section 5 with a debate on its consequences.

## 2. Classical and Quantum Games

The goal of game theory is to analyze decision-making systems involving two or more players cooperating or not with each other. An important feature in games is that the reward one player receives depends not only on the action she chooses but also on other players actions. It is well known that a game is defined by three elements: players, strategies, and rewards. This work is based on two players' games with two pure strategies each. However, players are also allowed to use mixed strategies where players can assign probabilities to each pure strategy and then randomly select between them.

Since probabilities are continuous, there are infinitely many mixed strategies available to a player. Then, rewards are defined by a 2 × 2 payoff matrix. In Table 1, it is possible to observe a general representation of a 2 × 2 payoff matrix (where [a,c,e,g] and [b,d,f,h] are the rewards of players 0 and 1, respectively). The following tables represent all the games that will be studied in the rest of the article (Prisoner's Dilemma Table 2, Deadlock Table 3, Discoordination Table 4a and Selfish game Table 4b), where player 0 can select between row actions and player 1 between column actions and obtain a reward of $(R_{player0}, R_{player1})$.

**Table 1.** General matrix representation of a game with two players and two strategies.

| \ | Player 1 | | |
|---|---|---|---|
| \ | \ | C | D |
| Player 0 | C | (a; b) | (c; d) |
| | D | (e; f) | (g; h) |

**Table 2.** Prisoner's Dilemma payoff matrix representation (e > a > g > c and d > b > h > f).

| (**a**) Version 1 | | | |
|---|---|---|---|
| \ | Player 1 | | |
| \ | \ | C | D |
| Player 0 | C | (6.6; 6.6) | (0; 10) |
| | D | (10; 0) | (3.3; 3.3) |

| (**b**) Version 2 | | | |
|---|---|---|---|
| \ | Player 1 | | |
| \ | \ | C | D |
| Player 0 | C | (5; 5) | (−10; 30) |
| | D | (30; −10) | (−5; −5) |

**Table 3.** Deadlock game payoff matrix representation (c > a > g > e and f > b > h > d).

| (**a**) Version 1 | | | |
|---|---|---|---|
| \ | Player 1 | | |
| \ | \ | C | D |
| Player 0 | C | (6.6; 6.6) | (10; 0) |
| | D | (0; 10) | (3.3; 3.3) |

| (**b**) Version 2 | | | |
|---|---|---|---|
| \ | Player 1 | | |
| \ | \ | C | D |
| Player 0 | C | (5; 5) | (30; −10) |
| | D | (−10; 30) | (−5; −5) |

To study quantum games we follow the *EWL* [1] protocol for 2 players. The first step is to assign a quantum state to each of the possible strategies. In the case of two strategies, for example, in the Prisoner's Dilemma, *cooperate* → $|0\rangle$ and *defect* → $|1\rangle$. The second step is to create a quantum circuit where each player is assigned a qubit that starts in state $|0\rangle$. The third step is to create an entangled state between all the players. This is done by applying the entangling operator $J = cos(\frac{\gamma}{2}) * \mathbb{I}^{\otimes N} + i * sin(\frac{\gamma}{2}) * \sigma_x^{\otimes N}$, as seen in Figure 1, where $\mathbb{I}$ is the identity matrix, $\sigma_x$ the Pauli X-gate, $N = 2$ represents the number of players and $\gamma$ a value determining the amount of entanglement, $\gamma = 0$ being no entanglement at all and $\gamma = \frac{\pi}{2}$ maximum entanglement.

**Table 4.** Payoff matrix representation of other useful games.

**(a)** Discoordination game.

| \ | | Player 1 | |
|---|---|---|---|
| | \ | R | L |
| Player 0 | R | (10; 0) | (0; 10) |
| | L | (0; 10) | (10; 0) |

**(b)** Selfish game.

| \ | | Player 1 | |
|---|---|---|---|
| | \ | R | L |
| Player 0 | R | (0; 0) | (0; 10) |
| | L | (10; 0) | (0; 0) |

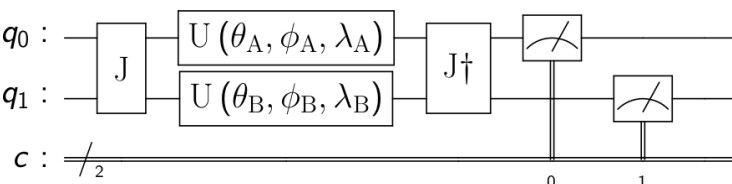

**Figure 1.** EWL game model for 2 players. Where $q_0$ and $q_1$ are the initial quantum states of the players and $c$ is a classical register where the qubits measurements are stored.

In the fourth step, every player chooses her most suitable strategy individually and independently. This is done by modifying the state of her own qubit locally. To do this, every player applies one or more one-qubit gates, modifying the state of her qubit. A general one-qubit gate [28] is a unitary matrix that can be represented as:

$$U(\theta, \phi, \lambda) = \begin{pmatrix} \cos(\frac{\theta}{2}) & -e^{i\lambda}\sin(\frac{\theta}{2}) \\ e^{i\phi}\sin(\frac{\theta}{2}) & e^{i(\phi+\lambda)}\cos(\frac{\theta}{2}) \end{pmatrix} \tag{1}$$

We can already highlight the fact that while classic players have only 2 possible pure strategies (e.g., cooperate or defect), quantum players have an infinite number of pure strategies, that is, any combination of real value for the three parameters $\theta$, $\phi$, and $\lambda$. The fifth step is to apply the operator $J^\dagger$ ($J$ conjugate transpose) after the players' strategies. Finally, the sixth step consists of measuring the state of the qubits to read the classical outputs of the circuit and, therefore, the final action of each player. The readouts are used as inputs of the payoff matrix to determine the players' rewards.

One last thing to add is the fact that we are going to replace the three parameters general one-qubit gate $U(\theta, \phi, \lambda)$ by three one-parameter rotation one-qubit gates $R_X(\varphi_1)R_Y(\varphi_2)R_X(\varphi_3)$, with $R_X(\varphi) = exp(-i\frac{\varphi}{2}X) = \begin{pmatrix} \cos(\frac{\varphi}{2}) & -i\sin(\frac{\varphi}{2}) \\ -i\sin(\frac{\varphi}{2}) & \cos(\frac{\varphi}{2}) \end{pmatrix}$ and $R_Y(\varphi) = exp(-i\frac{\varphi}{2}Y) = \begin{pmatrix} \cos(\frac{\varphi}{2}) & -\sin(\frac{\varphi}{2}) \\ \sin(\frac{\varphi}{2}) & \cos(\frac{\varphi}{2}) \end{pmatrix}$. This is possible without losing generality since $U(\theta, \phi, \lambda) = e^{i\alpha}R_{\hat{n}}(\beta)R_{\hat{m}}(\gamma)R_{\hat{n}}(\delta)$ [28]. Having said that, the circuit from Figure 1 becomes the one from Figure 2.

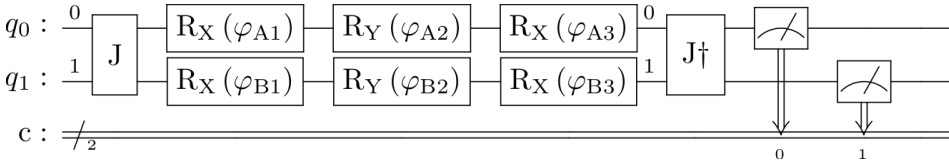

**Figure 2.** EWL game model for 2 players with rotation gates.

### 3. Learning Model

This section presents a new method for learning mixed strategies in quantum games. The results are focused on games with two players but the method is extensible for N players. The main difficulty when calculating mixed strategies in quantum games is that the possible pure strategies are already infinite (any real value for $\theta$, $\phi$, and $\lambda$). Therefore, a mixed strategy is obtained through a probability density function over these three continuous variables.

*3.1. Learning Algorithm*

First, the values of $\theta$, $\phi$, and $\lambda$ are restricted to any real value between $[0; 2\pi)$ (since they are angles). So, the final mixed strategy is obtained through a probability density function (PDF) of three variables. The principal idea is to use Q-learning [29], discretizing the strategy space, to learn the approximate value of every strategy, and construct a PDF with it. Players select their strategies at each iteration by sampling from the PDF. They receive a reward, update their own Q-tables and PDFs with this new information, and continue with the following iteration.

Three difficulties arise when agents try to learn their mixed strategies in quantum games. The first problem arises because when learning in games, each player's reward depends not only on the strategy she chooses but also on the strategies of other players. In other words, the feedback that a player uses to update the Q-value of a strategy will vary depending on the strategies that the other players have selected. This multi-agent scenario makes learning more difficult than the single-agent learning case where the feedback depends only on the strategy selected by the single player.

The second problem comes from the fact that there is a difference in nature between the classical and quantum strategies. Although pure classical strategies return always deterministic outcomes and mixed classical strategies always lead to random outcomes, pure quantum strategies can lead to random outcomes. For example, in Prisoner's Dilemma version 2a, if player0 selects the strategy $S_0 = (\theta_0, \phi_0, \lambda_0) = (\pi, 0, 0)$ and player1 also selects the strategy $S_1 = (\theta_1, \phi_1, \lambda_1) = (\pi, 0, 0)$, the quantum state before measuring is going to be $|\psi_{out}\rangle = J^\dagger(U(\pi, 0, 0) \otimes U(\pi, 0, 0))J|00\rangle = \frac{|00\rangle + |01\rangle + |10\rangle + |11\rangle}{2}$. This means that, the final actions are going to be (cooperate, cooperate) with probability 0.25, (cooperate, defect) with probability 0.25, (defect, cooperate) with probability 0.25, and (defect, defect) with probability 0.25. Therefore, even if both players repeat their strategies, rewards for player0 (and player1) will be 6.6 with probability 0.25, 0 with probability 0.25, 10 with probability 0.25, and 3.3 with probability 0.25. This nondeterministic property of pure quantum strategies makes learning of mixed quantum strategies slower.

The third one is due to the fact that when dealing with mixed strategies, learning the Q-values is not enough. Agents need to be able to properly transform that information into a probability density function that makes them neither too greedy players nor too hesitant. A greedy behavior will make agents try to obtain more rewards by using the present estimated value of the strategies and not learning enough from the environment, this concept is called exploitation and may cause convergence to local minima. On the other hand, A hesitant behavior will cause agents to primarily focus on improving their knowledge about each action instead of obtaining more rewards, this is called exploration and may cause no convergence at all.

Having said that, players start in the first iteration by initializing their Q-table, ideally this table will converge to a quantitative representation of how good each strategy is. Then, they initialize in zero a counter that indicates how many times each strategy was selected. This counter is going to be used to update the Q-table, the more times a strategy has been selected by the player, the lower the influence that the obtained reward will have when updating its Q-value. Finally, they select a strategy at random. After all the players have a strategy, they play the game and obtain the corresponding reward according to its $2 \times 2$ payoff matrix.

In the next iteration, agents use their selected strategies and received rewards to update the counter: $count(s)_t = count(s)_{t-1} + 1$, and the Q-value: $Q_t(s) = Q_{t-1}(s) + (\frac{1}{count_t(s)}) * (R_{t-1} - Q_{t-1}(s))$. Now, players have to select a new strategy. In order to achieve this, they are going to convert Q-values into probabilities and create a PDF from which they are going to sample to select the next strategy. The probability for each strategy is going to be defined by the softmax function: $p_t(s) = \frac{e^{\frac{Q_t(s)}{T}}}{\sum_{\text{all s}} e^{\frac{Q_t(s)}{T}}}$, where $T$ is called a temperature parameter. For high temperatures ($T \to \infty$), all strategies have nearly the same probability and the lower the temperature, the more Q-values affect the probability. For a low temperature ($T \to 0$), the probability of the strategy with the highest expected reward tends to 1.

There are two last concepts worth mentioning before going on to study the results of the algorithm. To avoid local optima, we add an $\epsilon$-greedy factor. In every iteration, each player generates a random real number between 0 and 1 and if this number is less than $\epsilon$, the player selects its strategy randomly instead of sampling from the PDF. Lastly, the $T$ factor of the softmax function will start from a initial high value $T_o$ (prioritizing exploration) and then decrease to a final low value $T_f$ (prioritizing exploitation) following the equation: $T = T_f + (T_o - T_f)e^{-\frac{t}{\tau}}$, $\tau$ being a decay rate factor.

In Algorithm 1, it is possible to observe a complete description of the learning method by using pseudocode.

---

**Algorithm 1** Agents learning mixed strategies in quantum games

---

**Require:** $N \geq 2$
  $N \leftarrow 2$          ▷ Number of players
  $T_o \leftarrow 4$          ▷ Initial Temperature
  $T_f \leftarrow 0.125$          ▷ Final Temperature
  $\tau \leftarrow 30000$          ▷ Velocity of T decay
  $\epsilon \leftarrow 0.01$          ▷ $\epsilon$-greedy factor
  **for** $t = 0$ **to** 200000 **do**          ▷ Number of iterations
    $T \leftarrow T_f + (T_o - T_f)e^{-\frac{t}{\tau}}$          ▷ Update Temperature
    **for** $n = 0$ **to** $N - 1$ **do**          ▷ For all players
      **if** $t = 0$ **then**          ▷ First step
        $count_n \leftarrow 0$          ▷ Initialize strategy counter
        $Q_n \leftarrow Q_0$          ▷ Initialize Q-table
        $strategy[n] \leftarrow strategies(randint)$      ▷ Return random strategy for player n
      **else**
        $count_n(s) \leftarrow count_n(s) + 1$      ▷ Update counter for strategy s
        $Q_n(s) \leftarrow Q_n(s) + (R[n] - Q_n(s))/count_n(s)$    ▷ Update Q-table for strategy s
        **if** random(1) $< \epsilon$ **then**          ▷ If not greedy
          $strategy[n] \leftarrow strategies(randint)$      ▷ Return random action for player n
        **end if**
        **for** s **in** strategies **do**          ▷ For all strategies
          $p(s) \leftarrow \frac{e^{\frac{Q(s)}{T}}}{\sum_s e^{\frac{Q(s)}{T}}}$      ▷ Update probability density function over strategies
        **end for**
        str $\leftarrow$ sample from p      ▷ Select a strategy by sampling in the PDF
        $strategy[n] \leftarrow strategies(str)$      ▷ Return the selected strategy for player n
      **end if**
    **end for**
    $R \leftarrow GAME(strategy)$      ▷ Obtain the Reward for all players from their strategies
  **end for**

---

### 3.2. Decentralized Model

The learning system is decentralized, each agent makes local autonomous decisions towards its individual goals which may possibly conflict with those of other agents. They

will learn individually and independently, without communication between them or a central ordering influence of a centralized system that exercises control over the lower-level components of the system directly.

In addition, agents will not have perfect information about the environment. Specifically, agents will not know the payout matrix of the game they are currently playing, they will not know the strategy or reward other agents are receiving at any point in time, and they do not even know how many agents they are playing against. Their *only* feedback is the reward they receive after selecting a strategy.

Both the decentralized and the imperfect information model for two players can be observed in Figure 3. Players select a strategy $S_X = [\varphi_{X1}, \varphi_{X2}, \varphi_{X3}]$ and send it to the quantum device. The two strategies of the players work as 6 parameters ($\varphi_{A1}$, $\varphi_{A2}$, $\varphi_{A3}$, $\varphi_{B1}$, $\varphi_{B2}$, and $\varphi_{B3}$) in a parameterized quantum circuit. The quantum circuit is executed and the readouts are mapped to their corresponding classical actions (the ones corresponding to the first step of the *EWL* protocol). These actions are used to play the ongoing game and the rewards are sent to their players. Players use their reward to adjust their strategy (according to Algorithm 1), select a new strategy and the parameterized quantum circuit is executed again.

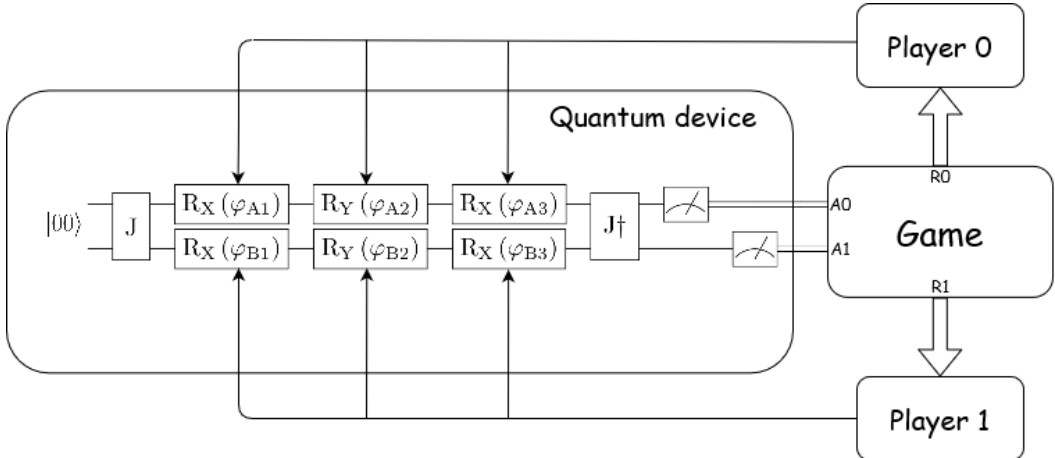

**Figure 3.** Model of two players learning mixed strategies using parameterized quantum circuits.

### 4. Results

In this section, we are going to start by applying Algorithm 1 to the classical setting of the games presented in Section 2 in order to verify its behavior. The equilibriums obtained are going to be compared with the well-known Nash equilibriums of these games. Later, Algorithm 1 will be applied to the quantum version of the games and check if the equilibriums changed and, if they did, if they did it for better or for worse.

In Section 4.1, quantum players will be maximally entangled at the time they apply their gates. However, in Section 4.2, we will study how entanglement influences the performance of quantum games. Similarly, in Section 4.1, quantum circuits are considered to be ideal regarding noise. In Section 4.3, noise is going to be introduced into the quantum circuits, in form of depolarizing noise, and we are going to characterize how this will end up biasing the behavior of players.

#### 4.1. Classic vs. Quantum Performance

It is also possible to find equilibriums of the classical version of the games from Section 2 by applying Algorithm 1. In this case, player only have two strategies (0 or 1, C or D, R or L, ...), so mixed strategies are defined by $p_0$ and $p_1$, where $p_0 + p_1 = 1$. In the classical equilibriums column from Table 5, it is clear that both the known Nash equilibriums and the equilibriums obtained by players applying Algorithm 1 approximately match.

**Table 5.** Classical and Quantum performance of games after applying Algorithm 1 (average rewards of the last 50,000 values).

| Games | Classical Equilibriums | | Quantum Equilibriums | |
|---|---|---|---|---|
| | Nash | Obtained | Tendency | Obtained |
| Prisoner's Dilemma v1 | [3.3; 3.3] | [3.316; 3.316] | [5; 5] | [4.968; 4.962] |
| Prisoner's Dilemma v2 | [−5; −5] | [−4.837; −4.857] | [10; 10] | [10.142; 9.618] |
| Deadlock game v1 | [6.6; 6.6] | [6.581; 6.585] | [5; 5] | [4.979; 4.964] |
| Deadlock game v2 | [5; 5] | [5.046; 5.055] | [10; 10] | [9.652; 9.464] |
| Disco-ordination Game | [5; 5] | [5.001; 4.999] | [5; 5] | [4.987; 5.013] |
| Selfish Game | [0; 0] | [0.148; 0.094] | [5; 5] | [4.952; 4.946] |

It is important to highlight the fact that all the players' strategies converge in pure strategies, with the exception of those who play the disco-ordination game ($p_0 = 0.5$ and $p_1 = 0.5$). This verifies the behavior predicted by known Nash equilibriums in all cases.

The (seemingly Nash) equilibrium rewards and the learning behavior of agents learning how to play quantum games using Algorithm 1 are shown in Table 5 and in Figure 4, respectively. All values in Table 5 correspond to the average of the last 50,000, that is, after most of the exploration phase has been done. It is possible to observe how in four (Tables 2a,b, 3b, and 4b) out of the six games, the quantum performance is higher that the classical one. In addition, there is one case (Table 4a) where the rewards are equal and one where the classical case (Table 3a) outperforms the quantum one. The average rewards plotted in Figure 4 were calculated by taking the mean value from a window of the last 50,000 rewards, when possible:

$$average\_reward(t) = \begin{cases} mean(rewards(0:t)) & \text{if } t < 50{,}000 \\ mean(rewards(t - 50{,}000:t)) & \text{if } t \geqslant 50{,}000 \end{cases}$$

It is interesting to note that in both versions of Prisoner's Dilemma, quantum agents somehow overcome the famous dilemma and obtain a higher reward than the classical Nash equilibrium ([5; 5] instead of [3.3; 3.3] and [10; 10] instead of [−5; −5]). Moreover, in Prisoner's Dilemma v2 (Table 2b), the outcome is even higher than the classical Pareto efficiency ([10; 10] instead of [5; 5]). This second condition will be met as long as $[\frac{c+e}{2}; \frac{d+f}{2}] > [a; b]$ in Table 1.

Similarly, in the deadlock game, which is a game that does not present any dilemma in its classical setting, it does in the quantum world. Classical players have a mutual cooperation strategy that is both Nash equilibrium and Pareto efficiency. However, when players switch to the quantum game, they lose that privilege. If condition $[\frac{c+e}{2}; \frac{d+f}{2}] > [a; b]$ is met, this is good news, as quantum players still manage to have a new equilibrium with an even higher reward than classical players ([10; 10] instead of [5; 5]). Otherwise, the rewards of the classical players will end up surpassing the quantum ones ([6.6; 6.6] versus [5; 5]).

The last two games are: the disco-ordination game and the selfish game. The former does not present any changes when quantized, what is in itself a particularity. The latter, on the other hand, is the game which presents the greatest difference. Classical players always converge to a pure strategy (L), which gives them the least possible reward ([g; h] = [0; 0]), while quantum players converge to a strategy that allows them to obtain the highest possible reward, $[\frac{c+e}{2}; \frac{d+f}{2}] = [5; 5]$.

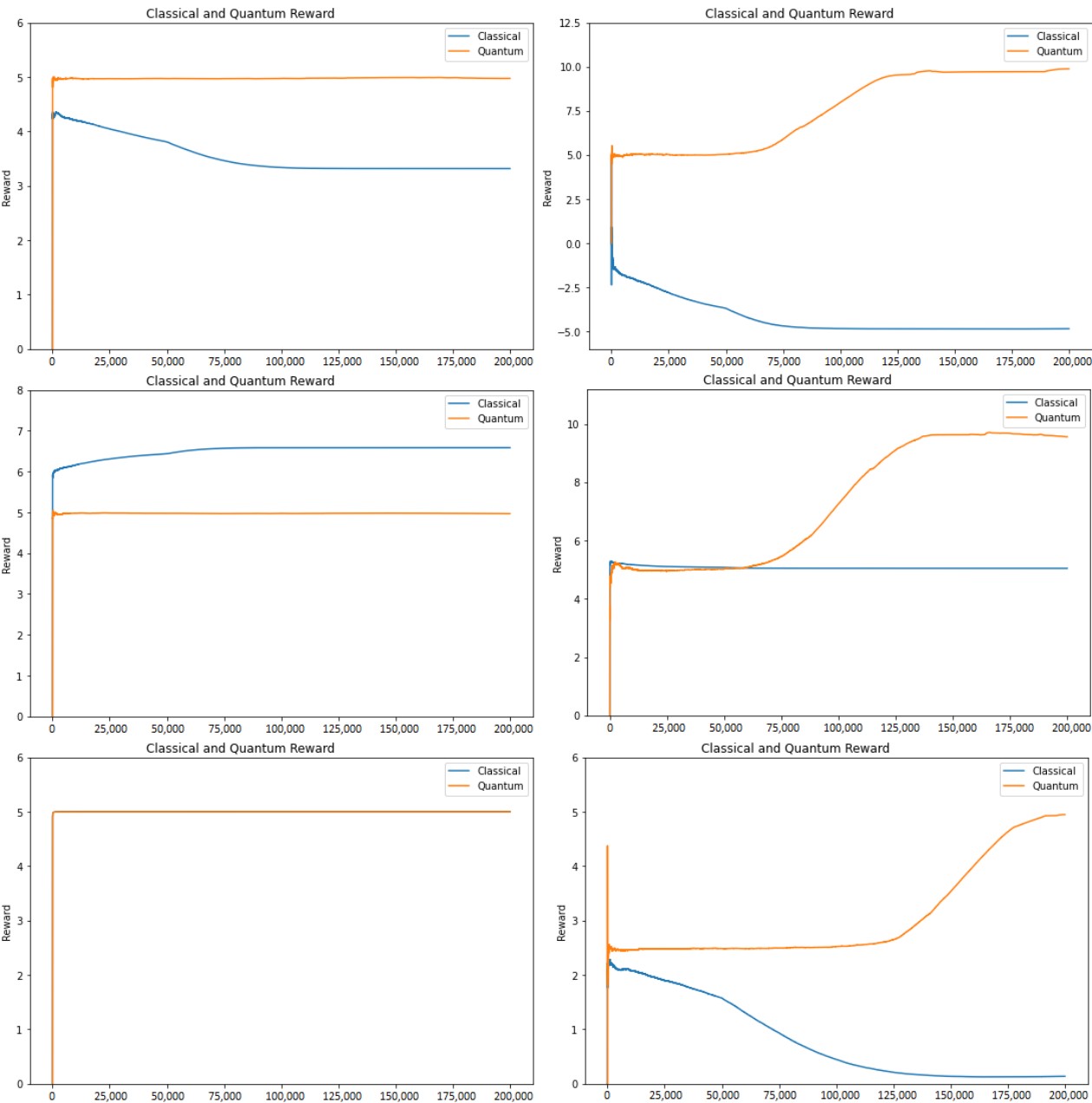

**Figure 4.** Average reward (from a window of the last 50,000 values) for agents learning how to play. Games from left to right. First row: Prisoner's Dilemma v1 and Prisoner's Dilemma v2. Second row: Deadlock game v1 and Deadlock game v2. Third row: disco-ordination game and selfish game.

### 4.2. Entanglement Dependency

So far, we have adopted a value of $\gamma = \frac{\pi}{2}$ in $J = cos(\frac{\gamma}{2}) * \mathbb{I}^{\otimes N} + i * sin(\frac{\gamma}{2}) * \sigma_x^{\otimes N}$. This assumes maximum entanglement between players. It is known that entanglement is the resource that allows advantages in quantum games over classical ones [30]. However, achieving this situation of maximum entanglement is sometimes costly to produce. Therefore, in Figure 5 it is possible to visualize how the ratio between quantum and classical rewards in equilibrium varies as a function of $\gamma$ ($\gamma = 0$ and $\gamma = \frac{\pi}{2}$ being no entanglement and maximum entanglement respectively) for three games with different behaviors.

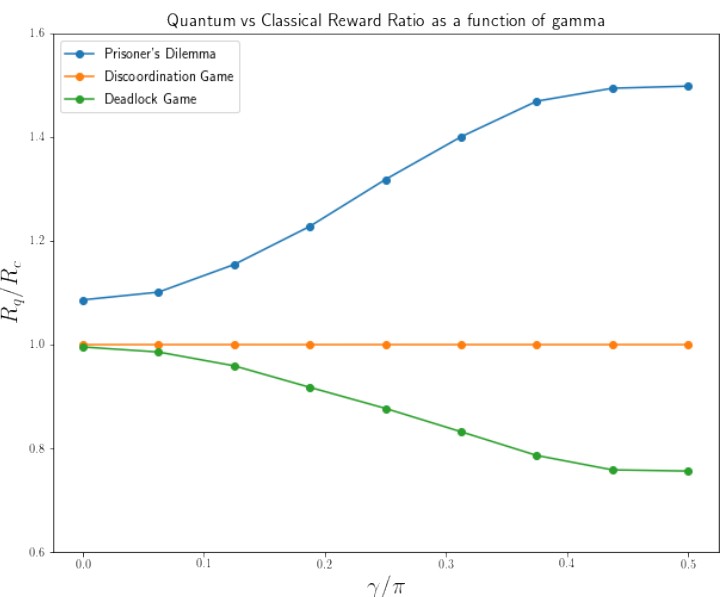

**Figure 5.** Quantum versus classical reward ratio as a function of the entanglement factor $\gamma$.

To begin with, when $\gamma = 0$, no entanglement, players of all quantum games have the same equilibriums than classical players, $\frac{R_q}{R_c} \simeq 1$. This behavior confirms that the differences between classical and quantum games come from entanglement and not from the fact that their strategy space was extended from two pure strategies to an infinite number of pure strategies. In the quantum game with $\gamma = 0$, players can still manipulate their qubits with any quantum gate from the infinite set of them, they have the same mixed strategy space with or without entanglement. Therefore, entanglement means quantum. Not entanglement means classical, even if they are allowed to manipulate qubits.

From then on, in Figure 5, as we increase the value of $\gamma$, we can see how the players' reward progressively approaches the previously calculated value with maximum entanglement. For Prisoner's Dilemma v1, $\frac{R_q}{R_c} = \frac{4.968+4.962}{3.316+3.316} \simeq 1.5$. For disco-ordination game, $\frac{R_q}{R_c} = \frac{4.987+5.013}{5.001+4.999} = 1$. For deadlock game v1, $\frac{R_q}{R_c} = \frac{4.979+4.964}{6.581+6.585} \simeq 0.75$. To conclude, we can say that in order to notice an advantage (or disadvantage) in quantum games, it is not necessary to exceed a certain entanglement level threshold. The greater the level of entanglement between the players, no matter how little it is, the greater the advantages (or disadvantages) players will experience.

### 4.3. Noise Dependency

A similar analysis but for a different phenomenon can be done by examining quantum noise. In Section 4.1, we assumed that, after the application of operator $J$, all players could apply their gates to their qubits in an ideal channel. This condition is also difficult to fulfill, so we will proceed to model this situation more carefully.

The noise model used will be the depolarizing channel. The state of the quantum system after this noise is: $\varepsilon(\rho) = \frac{\lambda I}{2} + (1 - \lambda)\rho = (1 - \lambda)\rho + \frac{\lambda}{3}(X\rho X + Y\rho Y + Z\rho Z)$, $\rho = |\psi\rangle\langle\psi|$ being the density matrix of the quantum state before noise is applied. The way to model this (Figure 6) is by adding a fourth gate after $R_X(\varphi_1)R_Y(\varphi_2)R_X(\varphi_3)$, this gate will be selected randomly following the probabilities:

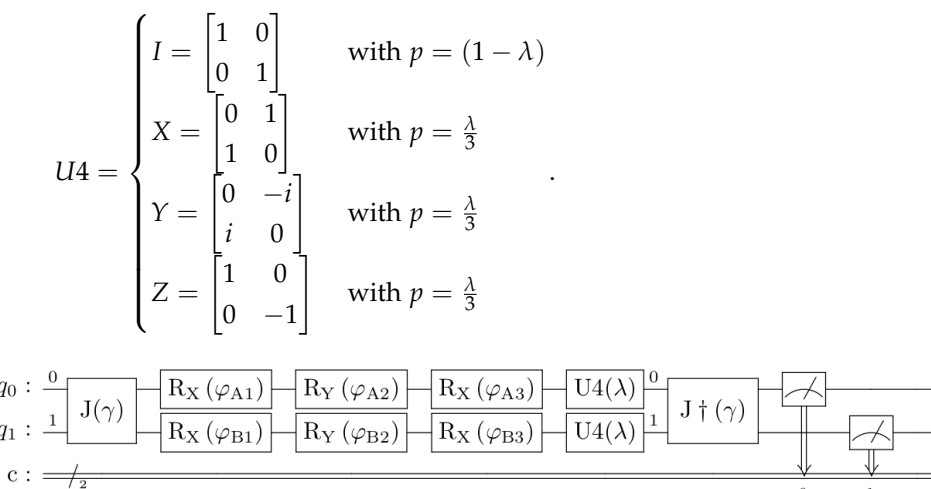

**Figure 6.** A complete model of the EWL protocol taking into account the entanglement factor and the depolarizing channel noise.

This means that the quantum state in each channel will remain intact with probability $(1 - \lambda)$ and will be modified with probability $\lambda$. To modify the quantum state of the channel, the X or Y or Z gates will be applied with the same probability. In Figure 7 it is possible to visualize how the performance of two games (Prisoner's Dilemma v2 and Selfish Game) varies as a function of $\lambda$.

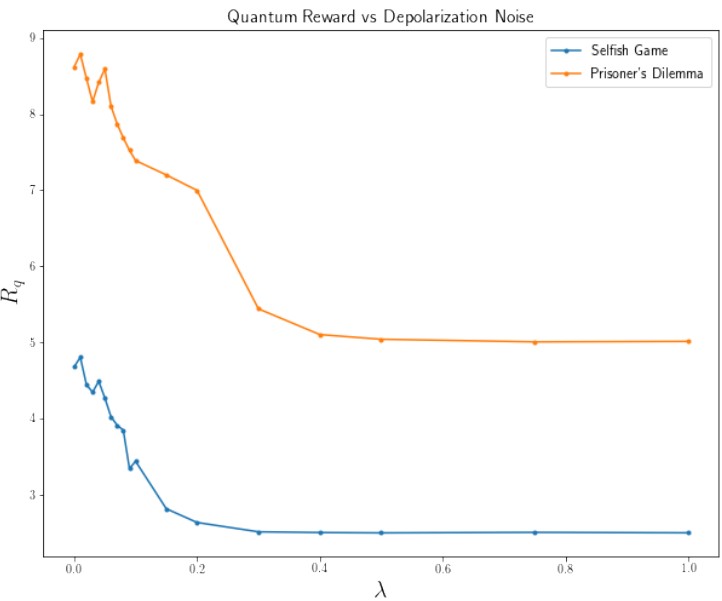

**Figure 7.** Reward of quantum players as a function of the $\lambda$ parameter for the depolarizing channel model.

In Figure 7, it is clear how noise affects the performance and the behavior of quantum players. The rewards begin to decrease rapidly even with small increases in the value of $\lambda$. In addition, for a value of $\lambda \geq 0.4$, the learning process of the players is so affected by noise that they can do no better than randomize between all the strategies. For Prisoner's Dilemma v2 (Table 2b), $R_q \simeq \frac{5 - 10 + 30 - 5}{4} = 5$. For selfish game (Table 4b), $R_q \simeq \frac{0 + 10 + 0 + 0}{4} = 2.5$.

One way to interpret this is that players are learning from a corrupt source. Sometimes, they select a strategy and receive and reward corresponding to another strategy. This causes players to behave irrationally, which, in turn, makes learning from the other player more

difficult. All this means that if a quantum system with little noise cannot be guaranteed, the learning of the players will be noticeably affected.

### 4.4. Mixed Strategies Visualization

Another way of understanding the behavior of players is by visualizing their strategies. Mixed strategies are represented by a probability density function over the strategy space. In quantum games the strategy space is represented by three continuous variables, each of them representing the angles of the quantum rotation gates $R_X(\varphi_1)$, $R_Y(\varphi_2)$, and $R_X(\varphi_3)$. In Figure 8, it is possible to observe different strategies learned by agents in quantum games under diverse settings. Each point in the PDF corresponds to a different strategy $S = (\varphi_1, \varphi_2, \varphi_3)$ and has assigned a value between 0 and 1 that corresponds to the probability of that strategy being selected by the player ($\sum_i P_{S_i} = 1$).

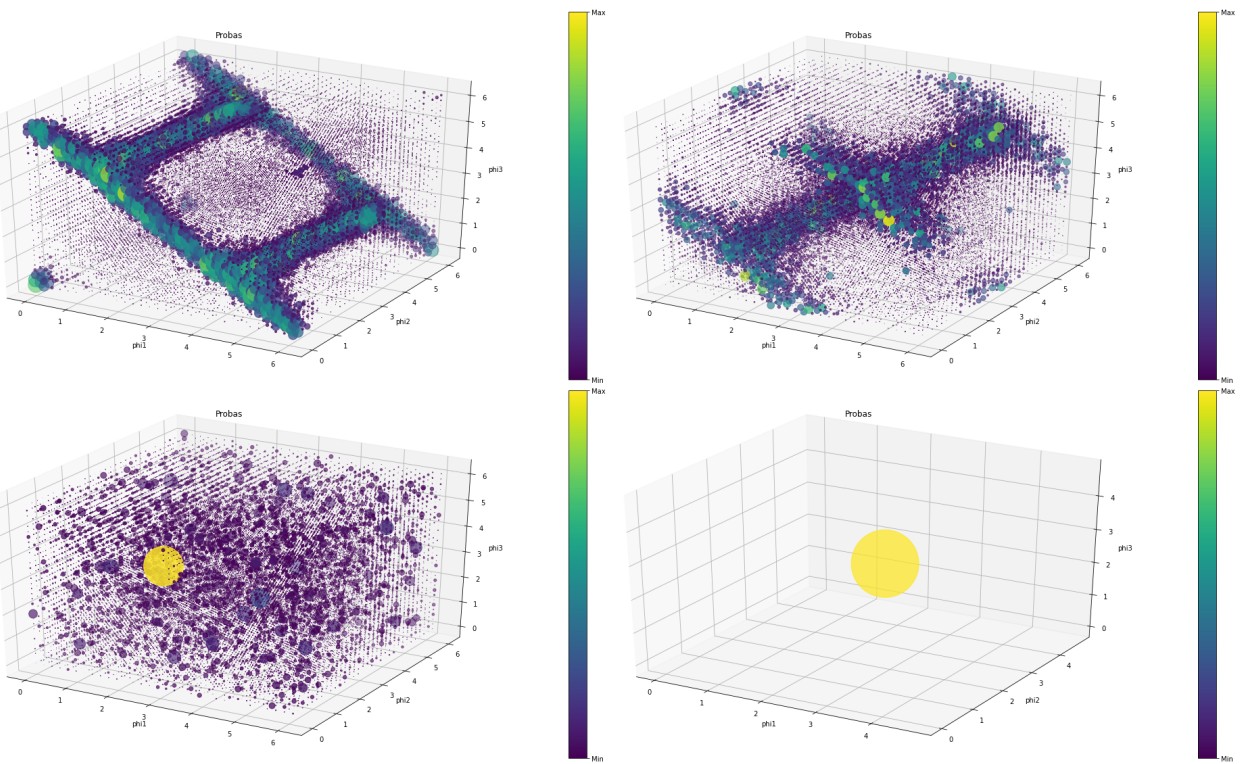

**Figure 8.** Probability Density Function over the three variables $(\varphi_1, \varphi_2, \varphi_3)$ representing different mixed quantum strategies. **Upper left**: deadlock game ($\gamma = 0$ and $\lambda = 0$). **Upper right**: Prisoner's Dilemma ($\gamma = 0$ and $\lambda = 0$). **Bottom left**: disco-ordination game ($\gamma = \frac{\pi}{2}$ and $\lambda = 0$). **Bottom right**: selfish game ($\gamma = \frac{\pi}{2}$ and $\lambda = 0$).

Each point is represented with a size and a color proportional to its value. The higher the value, the bigger its size and the yellower its color. It is possible to observe how, in some cases, mixed strategies converge to a pure strategy, $P_{S_i} = 1$ and $\forall j \neq i \, P_j = 0$ (Figure 8, bottom right). However, in most cases, mixed strategies converge on a particular distribution that may or may not have an interesting shape.

## 5. Conclusions

An algorithm for finding mixed strategies in quantum games was successfully designed. This algorithm has the property of being decentralized and allows players to learn in games with imperfect information. The proposed algorithm allows to systematically find equilibriums in all types of games and compare the rewards obtained in the quantum versions versus the classical ones.

The proper performance of the algorithm is verified by the matching of the obtained equilibriums for the classical version of the games with their corresponding Nash equilibriums. The use of the algorithm to find equilibriums on quantum games showed that the advantage (or disadvantage) quantum players have depends not only on the game they are playing but also on the relationships between the values of the payoff matrix.

The study of non-ideal scenarios to the proposed algorithm is accomplished by means of two parameters: $\gamma$ and $\lambda$, that represent the degree of entanglement between players and the quantum channel noise of each player, respectively. Moreover, we observe that in order to distinguish changes between classical and quantum games, some level of entanglement must exist, without which there is no difference with respect to classical games. Furthermore, it was observed that the outcomes of the quantum system are very sensible to noise and after a threshold ($\lambda \simeq 0.4$), the algorithm can no longer learn. Finally, the probability density functions obtained with the proposed algorithm for some of the quantum games studied are visualized.

In conclusion, this learning algorithm, despite the fact that players ignore most of the information related to the game (payoff matrix and other players actions), turned out to be flexible and efficient enough to pave the way to explore new ideas in the future about learning in quantum games, such as multiplayer arrangements or stochastic games. Finally, learning-based algorithms have proven useful enough in the classical world in the context of game theory. Our algorithm is a step towards extending its applications to the quantum.

**Author Contributions:** Conceptualization, A.S.; Formal analysis, A.S.; Investigation, A.S., O.G.Z., and C.M.A.; Project administration, C.M.A.; Software, A.S.; Supervision, O.G.Z. and C.M.A.; Validation, O.G.Z. and C.M.A.; Writing—original draft, A.S.; Writing—review and editing, O.G.Z. and C.M.A. All authors have read and agreed to the published version of the manuscript.

**Funding:** This research received no external funding.

**Institutional Review Board Statement:** Not applicable.

**Informed Consent Statement:** Not applicable.

**Data Availability Statement:** The data and scripts that support the findings of this study are available from the corresponding author upon reasonable request.

**Conflicts of Interest:** The authors declare no conflict of interest.

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
