# Peer review of "Learning Mixed Strategies in Quantum Games with Imperfect Information"

_quantumrep, doi:10.3390/quantum4040033_

Round 1
Reviewer 1 Report
The design of an algorithm for finding mixed strategies in quantum games seems to be interesting. The algorithm and the numerical simulations are useful. I would recommend this paper for publication.
Author Response
Author's Reply to the Review Report (Reviewer 1):
The design of an algorithm for finding mixed strategies in quantum games seems to be interesting. The algorithm and the numerical simulations are useful. I would recommend this paper for publication.
- We thank the reviewer for the recommendation and appreciation of our effort and our manuscript.
Reviewer 2 Report
The article may contain some interesting results regarding the learning strategy but needs to be refined as many aspects are unclear. It is especially difficult to understand what exactly is the result of this work. In literature there are variety of learning models have been proposed, with different motivations. What is the basis of the algorithm proposed in the work?
The work also requires extending the literature review to include various motivations in learning algorithms. I think that their method can also be applied in the concept of the quantum market, where the strategies are supply and demand curves (Schrodinger equation). A broader discussion of the potential uses would be useful. Also with regard to recent results in the field of quantum approaches to modeling games and complex systems such as financial markets.
I would like to see the work after the authors' corrections. In its present form, it is difficult to judge whether the work contains really new and significant results.
Author Response
Author's Reply to the Review Report (Reviewer 2):
The article may contain some interesting results regarding the learning strategy but needs to be refined as many aspects are unclear.
- Thank you very much for your words and we continue to detail our responses point by point.
It is especially difficult to understand what exactly is the result of this work.
- We modify the introduction by making explicit our contributions:
- “In terms of results, the main contributions of this work are: a) designing an algorithm to find equilibria in games with mixed quantum strategies, b) detecting that the equilibrium strategies not only depend on the game (e.g. prisoner's dilemma) but also on the relation between the values of the payoff matrix c) verifying that entanglement is the only resource that allows games to be quantum and d) characterizing the sensitivity of the algorithm to find equilibriums against noise in the quantum channels.”
In literature there are variety of learning models have been proposed, with different motivations. What is the basis of the algorithm proposed in the work?
- The introduction was rewritten by adding a comparison between our algorithm and other famous algorithms and justifying our approach.
- “There are already long-established learning strategies in the field such as fictitious play and no-regret learning. Fictitious play is a model-base algorithm where each player chooses the best response to their opponents' average strategy, therefore players need to know the actions selected by their opponents \cite{he2022finding}. No-regret learning, on the other hand, is a model-free approach that is based on the idea of minimizing the difference between the forecaster’s accumulated loss and that of an expert, so it requires players to know the reward function or payoff matrix. \cite{cesa2006prediction}. That said, we propose an algorithm to learn mixed strategies in quantum games where agents only have access to one feedback reward signal. Agents adjust their strategies without information about the payoff matrix of the game they are playing, the strategies selected by their opponents nor their rewards at each step, and even how many players they are playing against.”
- However, as far as our knowledge, we did not find any scientific article studying the consequences of applying learning algorithms in quantum games that allow players to use mixed quantum strategies. If the reviewer has any references to suggest, we would like to add them to our work.
The work also requires extending the literature review to include various motivations in learning algorithms. I think that their method can also be applied in the concept of the quantum market, where the strategies are supply and demand curves (Schrodinger equation). A broader discussion of the potential uses would be useful. Also with regard to recent results in the field of quantum approaches to modeling games and complex systems such as financial markets.
- Thank you very much for this suggestion, we add a paragraph explaining motivations for learning algorithms, especially when applied to quantum games:
- “Finally, the same motivations that drove the investigation of reinforcement learning algorithms in situations modeled by classical game theory, where players are unaware of the complete structure of the game, such as finance \cite{piotrowski2010reinforced} and network routing \cite{mammeri2019reinforcement}, encourage us to propose this algorithm for learning in quantum games. If the future of quantum computing allows us to create networks where quantum markets and the quantum internet become a reality, decentralized algorithms which support working with mixed quantum strategy and incomplete information will be absolutely useful for individuals to make the most out of the advantages of quantum systems \cite{khan2021quantum}.”
- Regarding the work “Schrödinger type equation for subjective identification of supply and demand”, even though we believe the is a really interesting work and we will consider it for future research, we could not find a direct way to use our algorithm. In that work, strategies are quantum states (following the Schrödinger equation) while in our work strategies are quantum gates (following the EWL protocol).
- We also added another potential application of our algorithm we found useful to explore in the near future:
- “The algorithm could also be easily adapted to be applied in systems where it is necessary to learn across games, that is, in situations where agents are playing different games that share an equivalent structure at the same time \cite{mengel2012learning}.”
I would like to see the work after the authors' corrections. In its present form, it is difficult to judge whether the work contains really new and significant results.
- We really appreciate the reviewer's suggestion and we believe that our work has improved significantly since the last version and hope that it will be accepted for publication.
Round 2
Reviewer 2 Report
The authors took my comments into account. In my opinion, the work is acceptable in its current form.